# Nimodipine-Dependent Protection of Schwann Cells, Astrocytes and Neuronal Cells from Osmotic, Oxidative and Heat Stress Is Associated with the Activation of AKT and CREB

**DOI:** 10.3390/ijms20184578

**Published:** 2019-09-16

**Authors:** Sandra Leisz, Sebastian Simmermacher, Julian Prell, Christian Strauss, Christian Scheller

**Affiliations:** Department of Neurosurgery, Medical Faculty, Martin Luther University Halle-Wittenberg, Ernst-Grube-Str. 40, 06120 Halle (Saale), Germany; Sebastian.simmermacher@uk-halle.de (S.S.); Julian.prell@uk-halle.de (J.P.); Christian.strauss@uk-halle.de (C.S.); Christian.scheller@uk-halle.de (C.S.)

**Keywords:** nimodipine, neuroprotection, astrocytes, neurons, Schwann cells, apoptosis

## Abstract

Clinical and experimental data assumed a neuroprotective effect of the calcium channel blocker nimodipine. However, it has not been proven which neuronal or glial cell types are affected by nimodipine and which mechanisms underlie these neuroprotective effects. Therefore, the aim of this study was to investigate the influence of nimodipine treatment on the in vitro neurotoxicity of different cell types in various stress models and to identify the associated molecular mechanisms. Therefore, cell lines from Schwann cells, neuronal cells and astrocytes were pretreated for 24 h with nimodipine and incubated under stress conditions such as osmotic, oxidative and heat stress. The cytotoxicity was measured via the lactate dehydrogenase (LDH) activity of cell culture supernatant. As a result, the nimodipine treatment led to a statistically significantly reduced cytotoxicity in Schwann cells and neurons during osmotic (*p* ≤ 0.01), oxidative (*p* ≤ 0.001) and heat stress (*p* ≤ 0.05), when compared to the vehicle. The cytotoxicity of astrocytes was nimodipine-dependently reduced during osmotic (*p* ≤ 0.01), oxidative (*p* ≤ 0.001) and heat stress (not significant). Moreover, a decreased caspase activity as well as an increased proteinkinase B (AKT) and cyclic adenosine monophosphate response element-binding protein (CREB) phosphorylation could be observed after the nimodipine treatment under different stress conditions. These results demonstrate a cell type-independent neuroprotective effect of the prophylactic nimodipine treatment, which is associated with the prevention of stress-dependent apoptosis through the activation of CREB and AKT signaling pathways and the reduction of caspase 3 activity.

## 1. Introduction

The calcium antagonist nimodipine binds to the α1-subunit of calcium channels and regulates the calcium influx. Nimodipine treatment is used to reduce vasospasm in relation to the management of aneurysmal subarachnoid hemorrhages (aSAHs) as well as having shown a positive effect on the outcome and the prevention of accompanied delayed ischemic neurological deficits (DIND) [1,2]. Furthermore, a neuroprotective effect after aSAH has been shown [3]. In vestibular schwannoma, laryngeal and maxillofacial surgery evidence suggests a beneficial effect of nimodipine [4,5,6,7,8,9]. Neuroprotection through nimodipine pretreatment has been demonstrated in vitro in PC-12 pheochromocytoma cells [10]. We have shown in previous studies that the cell death of neuroblastoma cells was significantly reduced after pretreatment with nimodipine prior to stress induction [11,12]. In addition, an upregulation of fatty acid 2-hydrolase (FA2H) was shown via nimodipine and stress induction on the mRNA and protein levels. FA2H is involved in the synthesis of 2-hydroxy fatty acids, which are incorporated in 2-hydroxy galactolipids. The molecular mechanism of the nimodipine neuroprotective effect is unclear. In mice hippocampal cells, an increased activation of the tyrosine receptor kinase B (TrkB) receptor, proteinkinase B (AKT) and cyclic adenosine monophosphate response element-binding protein (CREB) was shown [13]. Apart from these studies, the effect of nimodipine on various cell types of the nervous system has not been studied. To understand nimodipine’s neuroprotection mode of action, it is now mandatory to analyze which cells are affected. Therefore, the present study analyzed the influence of nimodipine on Schwann cell, neuronal cell and astrocyte stress-induced cell death and the associated molecular mechanisms.

## 2. Results

### 2.1. Nimodipine Decreases Cytotoxicity of Schwann Cells, Neuronal Cells and Astrocytes under Different Stress Conditions

The cytotoxicity was measured via lactate dehydrogenase (LDH) activity in cell culture supernatants after the treatment of cells with 150 mM NaCl (osmotic stress), 2% EtOH (oxidative stress) and incubation for 6 h at 42 °C (heat stress). The Schwann cells showed a nimodipine-dependent significant reduction of cytotoxicity up to 19% at osmotic stress (1 µM and 20 µM nimodipine, *p* ≤ 0.001; Figure 1A). Under oxidative stress, the cytotoxicity was reduced via the treatment of 1 µM, 10 µM and 20 µM nimodipine from 36% to 23%, 16% and 23% (*p* ≤ 0.001, Figure 1B), respectively. The analysis of the nimodipine-treated Schwann cells after heat stress demonstrated a reduction from 21% to 14% (10 µM nimodipine, *p* ≤ 0.01; Figure 1C) and 18% (20 µM nimodipine, *p* ≤ 0.01) in comparison to the vehicle cells. The treatment with 1 µM nimodipine showed no significant alteration of the Schwann cell cytotoxicity under heat stress.

The results of the neuron LDH release analysis demonstrated a decreased cytotoxicity of immortalized neurons at osmotic stress from 31% to 26% (Figure 2A; 10 µM nimodipine *p* ≤ 0.01) and 21% (20 µM, not significant in comparison to the vehicle control). The treatment with 1 µM nimodipine was not significantly altered, when compared to the vehicle. During oxidative stress, the cytotoxicity was reduced from 45% to 30% (Figure 2B; 10 µM and 20 µM nimodipine *p* ≤ 0.001) but only slightly under heat stress (Figure 2C; 10 µM nimodipine *p* ≤ 0.05) compared to the control cells (vehicle). The treatment with 1 µM nimodipine did not lead to a significant difference of neuronal cell toxicity under oxidative and heat-related stress.

During osmotic stress, the cytotoxicity of astrocytes was reduced from 25% to 13% (Figure 3A; 10 µM nimodipine *p* ≤ 0.01) and 14% (20 µM nimodipine *p* ≤ 0.05). After the incubation of the cells with 2% ethanol (oxidative stress), the results showed a decreased cell death of up to 25% (Figure 3B; 20 µM nimodipine *p* ≤ 0.001). Under heat stress, a reduction of the cytotoxicity of up to 17% (Figure 3C; not significant compared to the vehicle) was detected. The treatment with 1 µM nimodipine showed no significant reduction of the astrocyte cytotoxicity.

### 2.2. Caspase 3/7 Activity Analysis of Nimodipine Treated Schwann Cells, Neuronal Cells and Astrocytes

The influence of the nimodipine treatment on the caspase 3 and 7 activity of stressed cells was detected via Caspase-Glo 3/7 Assay (Promega). The analysis showed an alteration of caspase 3 and 7 activity through the nimodipine treatment under osmotic, oxidative and heat stress in Schwann cells (Figure 4A), neuronal cells (Figure 4B) and astrocytes (Figure 4C). Caspase 3 and 7 activity was statistically, significantly and nimodipine-dependently decreased to 83% during osmotic stress (10 µM nimodipine *p* ≤ 0.001), to approximately 60% during oxidative stress (10 µM and 20 µM nimodipine *p* ≤ 0.001) and to approximately 90% during heat stress (10 µM and 20 µM nimodipine *p* ≤ 0.001) for the Schwann cells compared to the vehicle cells. The neuronal cells showed a caspase activity reduction by 36% during osmotic stress (10 µM and 20 µM nimodipine *p* ≤ 0.001), to 82% and 70% during oxidative stress (10 µM and 20 µM nimodipine *p* ≤ 0.001) as well as to approximately 90% (*p* ≤ 0.001) during heat stress compared to the vehicle control. For the astrocytes, the caspase 3/7 activity was reduced to ~60% during osmotic (10 µM and 20 µM nimodipine *p* ≤ 0.001), to 71% during oxidative (20 µM nimodipine *p* ≤ 0.01) and to 75% during heat stress (10 µM nimodipine *p* ≤ 0.001) in comparison to the vehicle. The 20 µM nimodipine treatment led to an increased caspase 3/7 activity under heat stress in the astrocyte cell line (150%, *p* ≤ 0.001).

### 2.3. Neuroprotective Effect is Associated with Activation of AKT and CREB Signaling

The immunoblot analysis of the nimodipine-treated SW10 cells showed an increased AKT phosphorylation at serin residue 473 and CREB phosphorylation at serin residue 133 (Figure 5), which was detectable without stress and during the tested stress conditions. The amounts of total AKT, CREB and phospho-ATF-1 protein were not affected through the nimodipine treatment or stress incubation. The GAPDH protein levels served as the loading control.

## 3. Discussion

### 3.1. Nimodipine Pretreatment Leads to Cell Type-Independent Reduction of Cytotoxicity

Nimodipine is a well-tolerated drug with a good safety profile. Because of its lipophilic properties, nimodipine passes the blood-brain barrier in a better way than other calcium antagonists [14]. For cerebral infarction secondary to SAH, nimodipine is recommended as a first-line therapy. However, nimodipine improves the neurological outcome and reduces mortality [15], which is known to be separate from its vasodilatory effects [16].

In the last years, some clinical trials have shown a beneficial effect of nimodipine treatment for the preservation and regeneration of nerve function after resections of vestibular schwannoma and maxillofacial surgery [4,5,6]. Furthermore, medication of nimodipine prior to surgery was predominant compared to intraoperative initiation or no treatment [5,17,18,19]. In contrast, the *Very Early Nimodipine Use in Stroke* (VENUS) trial showed no beneficial effect of nimodipine treatment when nimodipine was applied 6 h after an acute stroke [20]. But different trials proved a reduced incidence and 40% better outcome of DIND following aSAH for nimodipine [21,22]. Mattsson et al. have shown the neuroprotective role of nimodipine as a cranial nerve protective agent in skull base surgery [23]. They demonstrated, via a rat intracranial facial nerve crush model and a nimodipine pretreatment three days prior to surgery, that nimodipine accelerates the time of functional nerve recovery [24]. Furthermore, nimodipine pretreatment reduces the incidence of postoperative cognitive dysfunction by decreasing hippocampal apoptosis in rats [25]. These data indicate that the pretreatment or prophylactic administration of nimodipine is required for its neuroprotective effect. Our data have shown that there is a neuroprotective effect as a result of a 24-h pretreatment with nimodipine. The stress models were established in previous studies [11,12,26]. Ethanol treatment is able to induce oxidative stress through acetaldehyde metabolism, which inhibits mitochondrial respiration at the level of complex I (NADH-ubiquinone oxido-reductase) and the coupling site I of oxidative phosphorylation [27]. The production of reactive oxygen species (ROS) during oxidative stress is involved in the pathogenesis of neuroinflammatory and neurodegenerative diseases [28,29]. Osmotic stress leads to demyelination and cell apoptosis [30,31]. In 2014 and 2017, Herzfeld et al. demonstrated the anti-apoptotic effect of nimodipine on neuroblastoma cells during osmotic, oxidative, mechanic and heat stress. In the present study, LDH activities in cell culture supernatants were statistically significantly reduced between 11 and 66% in Schwann cells, neurons and astrocytes under different stress conditions (Figure 1, Figure 2 and Figure 3). However, the different cell types have shown different levels of stress induction and responded differently to lower or higher levels of nimodipine. Nimodipine pretreatment cannot completely prevent cell death but reduces it. The results have shown that the best protection against oxidative stress was determined under the conditions used. For heat stress, a slight or not significant reduction of cytotoxicity was detected in neuronal cells and astrocytes.

In previous studies, nimodipine concentrations between 1–20 µM were appropriate for achieving a neuroprotective effect in cell culture [11,12], which also applies to the present results. This suggests that the neuroprotective effects of nimodipine are not dose-dependent in the concentrations between 1–20 µM, but that the time of treatment seems to be critical. On the one hand, the cell type-independent anti-apoptotic effect of nimodipine indicates a ubiquitous molecular mode of action, which underlines the putative neuroprotective potential of nimodipine therapy for other diseases like multiple sclerosis, Alzheimer’s or Parkinson’s disease [32,33,34,35,36]. On the other hand, the impact of analyzed cell systems is limited by the influence of in vivo factors like the immune response, cell-cell interaction and the interaction of cells with the tissue micro-milieu. In 2017, Herzfeld et al. showed a nimodipine-dependent upregulation of the FA2H, which is involved in the myelination of Schwann cells and oligodendrocytes. Additionally, Kuerten et al. could demonstrate, via a mouse model of multiple sclerosis (MS), that nimodipine treatment promotes the remyelination of nerve fibers and induces microglia apoptosis [32]. Within the context of MS, further analyses should investigate the influence of nimodipine treatment on astrocyte activation and the formation of reactive gliosis.

### 3.2. Nimodipine-Dependent Apoptosis Prevention Is Accompanied by Reduction of Caspase 3 and 7 Activity and Activation of CREB and AKT

Caspase 3 and 7 are effector caspases, irreversibly inducing the apoptosis and doing so in a direct manner via the degradation of the cell membrane and of the cytoskeleton actin filaments [37,38,39,40]. In principle, caspase 3 and 7 can be activated through the cleavage of procaspases both by extrinsic and intrinsic apoptosis pathways [41]. In the current study, we show for the first time that nimodipine reduces the effector caspase 3 and 7 activity under different stress conditions (Figure 4). This finding reveals that nimodipine is involved in the last step of an anti-apoptotic pathway and that it prevents the early apoptosis. In contrast, the higher concentration (20 μM) of nimodipine led to an increased caspase 3/7 activity in astrocytes under heat stress (Figure 4C). Under these conditions, the results of the cytotoxicity analysis showed a reduction of the cell death that was not significant. Regarding these inconsistent data, further analyses should clear these effects to examine the impact of nimodipine in the cell death of heat stressed astrocytes.

The analysis of the cell signaling pathways of Schwann cells has shown a nimodipine-dependent increased AKT phosphorylation at serin 473 without stress and osmotic, oxidative or heat stress (Figure 5). The AKT pathway is characterized as the coordinator of cellular functions, containing cell cycle progression, proliferation and cell protection [42,43,44]. It has been demonstrated that the activation of the AKT signaling pathway reduces oxidative stress in neuronal cells [45] and could play a therapeutic role in neurodegenerative diseases [46,47]. In the central nervous system, AKT modifies the synaptic plasticity and promotes neural cell survival, neuroprotection and neuroregeneration [48]. AKT is known to induce CREB phosphorylation [49,50]. Furthermore, our studies have also shown a nimodipine-dependent higher CREB phosphorylation at serin 133 (S133) (Figure 5). The activation of CREB (S133) has been proven to be a key regulator in promoting survival and in preventing DNA damage in neuronal cells during oxidative stress and hypoxia [51,52]. Evidence suggests that CREB is involved in an active process of neuroprotection [52]. Furthermore, the expression of deacetylase Sirtuin 1 (SIRT1), which is known to be involved in neuronal survival, is regulated by activated CREB [53]. Stress induction like oxidative stress and heat leads to the inactivation of SIRT1, accompanied by protein misfolding, cell death and neurodegeneration [54]. Moreover, SIRT1 is able to deacetylate AKT, which increases the AKT activation [55]. Due to these facts, it seems to be possible that SIRT1 is a missing link in the nimodipine-dependent signaling pathway. Therefore, one should analyze whether nimodipine is a putative activator of SIRT1. In accordance with our results, Koskimäki et al. described an increased AKT and CREB phosphorylation in mouse hippocampal cells after nimodipine treatment [13]. Therefore, our data confirm the neuroprotective effect of nimodipine, which is accompanied by the reduction of the caspase 3 and 7 activity and the activation of CREB and AKT signaling (Figure 6). This effect could improve the neurological outcome in the management of skull-based lesions and various neurological diseases via a prophylactic nimodipine treatment.

As a voltage-dependent L-type calcium channel blocker, nimodipine regulates the calcium influx in the cells [56,57], which could prevent calcium overload causing apoptosis. In contrast, nifedipine, which is also a calcium channel blocker of the same class and which has a high structural identity with nimodipine, showed no or lower neuroprotective properties [12,58]. Therefore, further studies should clarify whether the shown anti-apoptotic mechanism is controlled by the prevention of calcium overload.

## 4. Materials and Methods

### 4.1. Cell Lines

The murine cell lines C8-D1A (CLR-254, astrocytes), RN33B (CLR-2825, neuronal cells) and SW10 (CLR-2766, Schwann cells) were obtained from the American Type Culture Collection (ATCC, Manassas, VA, USA). Astrocytes and Schwann cells were cultured in Dulbecco’s Modified Eagle’s Medium (DMEM; ThermoFisher Scientific, Darmstadt, Germany), supplemented with 10% fetal calf serum (FCS; Gibco, ThermoFisher Scientific), 2 mM glutamine (Biochrom AG, Merck, Darmstadt, Germany), 100 U/mL penicillin and 100 μg/mL streptomycin (ThermoFisher Scientific) with 5% CO_2_ at 37 °C. The neuronal cells were cultured in DMEM/F12 (1:1, ThermoFisher Scientific) containing the same supplements at 5% CO_2_ and 33 °C. The cell lines were routinely analyzed for mycoplasma contamination. All experiments were carried out in three independent biological replicates.

### 4.2. Nimodipine Treatment

1 × 10^4^ SW10 and 5 × 10^4^ C8-D1A or RN33B cells were seeded in 24-well plates and treated 24 h prior to the stress application with 1 µM, 10 µM and 20 µM nimodipine [11] (Bayer AG, Leverkusen, Germany), which was diluted in absolute ethanol (EtOH). Equal amounts of EtOH were added to the control cells (0.1% final concentration, vehicle). The nimodipine solutions and the treated cells were protected from light.

### 4.3. Stress Induction

Cytotoxicity was induced with 2% EtOH, 150 mM NaCl and a 6-h heat incubation at 42 °C, as described in Herzfeld et al. 2017 [12] and shown schematically in Figure 7.

### 4.4. Cytotoxicity Measurement

The induction of cell death was determined after 24 h through the LDH release of cells with the Cytotoxicity Detection Kit (Roche, Grenzach-Wyhlen, Germany), as previously described in detail [11]. For the calculation of the cell death rate, the absorbance of the culture medium (background control) was subtracted from the values. In order to normalize the data, the untreated cells were lysed with 2% Triton X-100, and the absorbance of total lysis was set to 100%. The wells were measured at four points. The diagrams show the means and standard deviations (SD) of triplicates from one representative assay out of three biological replicates.

### 4.5. Western Blot Analysis

24 h after the stress induction (Figure 1), the cells were washed and harvested with ice-cold PBS containing a protease and phosphatase inhibitor cocktail (ThermoFisher Scientific). The proteins were processed according to Laemmli, as previously described [59,60]. To summarize, proteins were separated by SDS PAGE, followed by blotting onto a nitrocellulose membrane (0.45 µm, Amersham, GE Healthcare, Freiburg, Germany). The membranes were incubated overnight at 4 °C with the primary antibodies pAKT (S473), AKT, pCREB (S133), CREB or GAPDH (all purchased from Cell Signaling, Frankfurt, Germany) after blocking with 5% skim milk (Carl Roth, Karlsruhe, Germany) diluted in Tris-buffered saline. The following day the immunoblots were developed using the ECL method (Pierce, ThermoFisher Scientific). Chemiluminescence signals were detected with a CCD camera (ImageQuant LAS4000, GE Healthcare).

### 4.6. Caspase 3/7 Assay

The activity of caspase 3 and 7 was measured using a Caspase-Glo 3/7 assay (Promega, Mannheim, Germany), according to the manufacturer’s instructions. 1 × 10^4^ cells were seeded in white 96-well cell culture plates (Greiner Bio-One, Frickenhausen, Germany) in triplicates and were pretreated for 24 h with 1, 10 and 20 μM nimodipine or ethanol alone (solvent control). 24 h after the stress induction with 150 mM NaCl (osmotic stress), 2% EtOH (oxidative stress) or incubated 6 h at 42 °C (heat stress), the cells were incubated for 30 min with 100 μL Caspase-Glo 3/7 reagent in the dark, followed by a measurement of the luminescence signals (Tecan Reader Infinite F200PRO, Tecan, Männedorf, Switzerland). A cell-free culture media served as the background control. Signals of untreated stressed cells were set to 100%. The results showed the means ± SD of three technical replicates.

### 4.7. Statistical Analysis

The statistical analysis was performed with a one-way ANOVA followed by a Tukey’s post hoc test (SPSS statistics 25, IBM, Ehningen, Germany). The significance was accepted if the *p* values were ≤0.05 (* *p* ≤ 0.05, ** *p* ≤ 0.01, *** *p* ≤ 0.001). The data were expressed as the mean ± S.D.

## 5. Conclusions

Nimodipine pretreatment leads to cell type-independent neuroprotection through the decreased activation of caspase 3 and 7 and the increased activation of CREB and AKT signaling (Figure 6). This cell type-independent neuroprotection offers nimodipine as a neuroprotective substance against a wide range of stress-induced neuronal damages during surgery or various neurodegenerative diseases. Prospective investigations of the anti-apoptotic mechanisms should be continued in vitro and verified in vivo, in order to ascertain whether these mechanisms can be transferred to patients’ tissues.

## Figures and Tables

**Figure 1 ijms-20-04578-f001:**
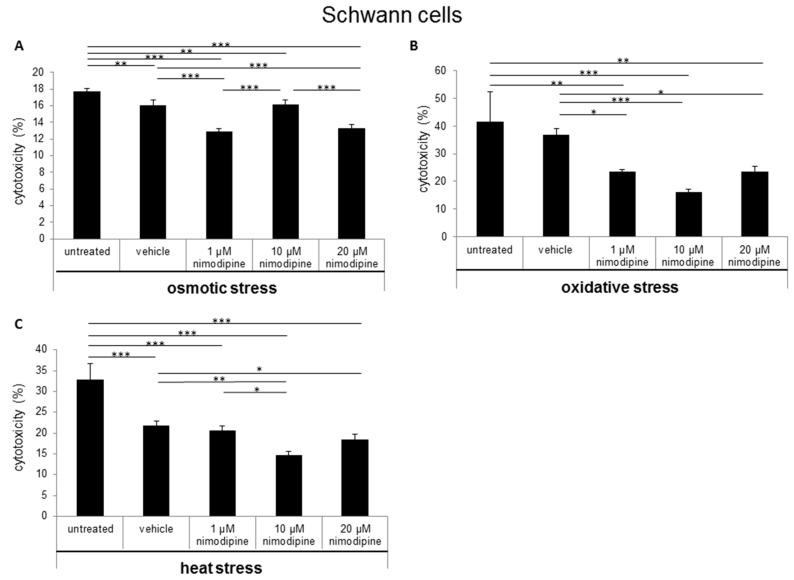
Nimodipine decreases the cytotoxicity of Schwann cells during different stress conditions. 5 × 10^4^ SW10 cells were seeded and pretreated with the vehicle, 1, 10 or 20 μM nimodipine. The untreated cells served as control cells. After 24 h, the cytotoxicity was induced with (**A**) 2% EtOH, (**B**) 150 mM NaCl or (**C**) a 6-h heat incubation at 42 °C. The lactate dehydrogenase LDH activity was measured from cell culture supernatants as described in the material and method section. A statistical analysis was performed via a one-way ANOVA, followed by a Tukey post hoc test. The significance was accepted if the p values were ≤ 0.05 (* *p* ≤ 0.05, ** *p* ≤ 0.01, *** *p* ≤ 0.001).

**Figure 2 ijms-20-04578-f002:**
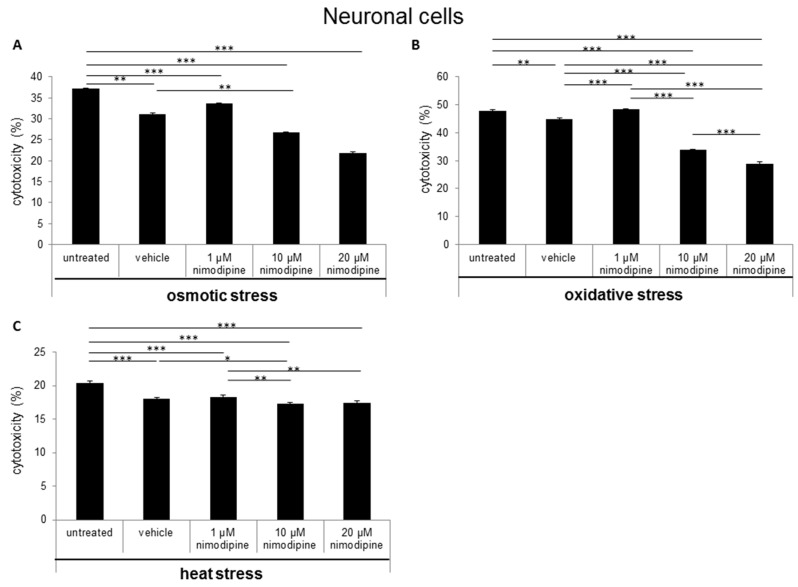
Nimodipine decreases the cytotoxicity of neuronal cells during different stress conditions. As described in the materials and methods section, 1.5 × 10^5^ RN33B cells were seeded and treated with the vehicle, 1, 10 or 20 μM nimodipine. After 24 h, the cytotoxicity was induced with (**A**) 2% EtOH, (**B**) 150 mM NaCl or (**C**) a 6-h heat incubation at 42 °C. The LDH activity was detected via the Cytotoxicity Detection Kit (Roche) according to the manufacturer’s instructions. The significance of the values was accepted if the p values were ≤ 0.05 (* *p* ≤ 0.05, ** *p* ≤ 0.01, *** *p* ≤ 0.001). The data were expressed as the mean ± S.D.

**Figure 3 ijms-20-04578-f003:**
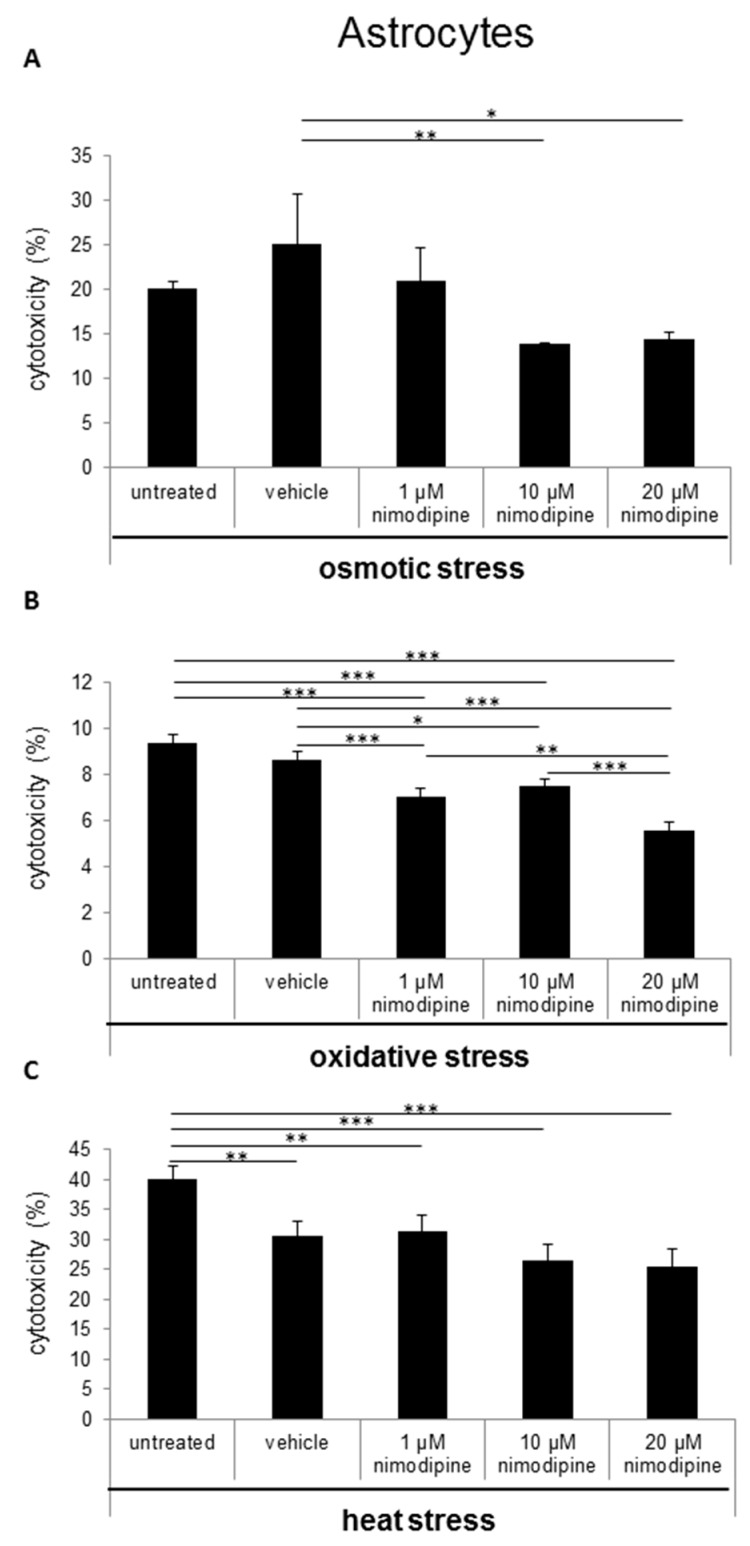
Nimodipine decreases the cytotoxicity of astrocytes during different stress conditions. 1 × 10^5^ C8-D1A cells were seeded in 24-well plates and treated for 24 h with the vehicle, 1, 10 or 20 μM nimodipine. The cytotoxicity was induced after 24 h with (**A**) 2% EtOH, (**B**) 150 mM NaCl or (**C**) 6 h of heat incubation at 42 °C. The LDH activity was analyzed from the cell culture supernatant after an additional 24 h and normalized to the total cell lysate values. The data of the cell free culture medium served as the background control. The significance was accepted if the p values were ≤ 0.05 (* *p* ≤ 0.05, ** *p* ≤ 0.01, *** *p* ≤ 0.001). The bars indicate the standard error mean.

**Figure 4 ijms-20-04578-f004:**
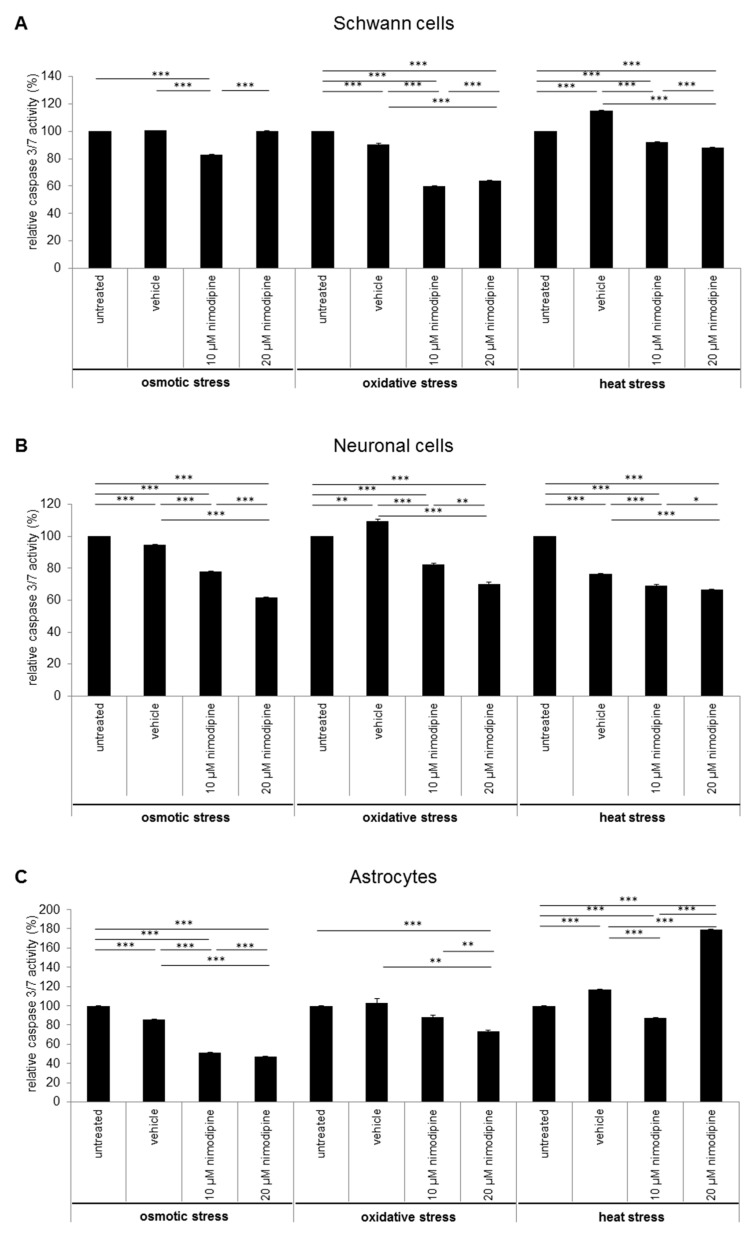
The reduced caspase 3/7 activity through the nimodipine treatment. 1 × 10^4^ cells were seeded in 96-well plates and treated as described in the method section. 24 h after stress induction, 100 μL Caspase-Glo substrate was added to the cells and incubated for 30 min without light, before the luminescence was measured. The figures show the means and standard derivations of the relative caspase 3 and 7 activity of the (**A**) SW10 Schwann cells, (**B**) RN33B neurons and (**C**) C8-D1A astrocytes after osmotic, oxidative and heat stress. The caspase activity of the control cells was set to 100%. The significance was accepted if the *p* values were ≤ 0.05 (* *p* ≤ 0.05, ** *p* ≤ 0.01, *** *p* ≤ 0.001).

**Figure 5 ijms-20-04578-f005:**
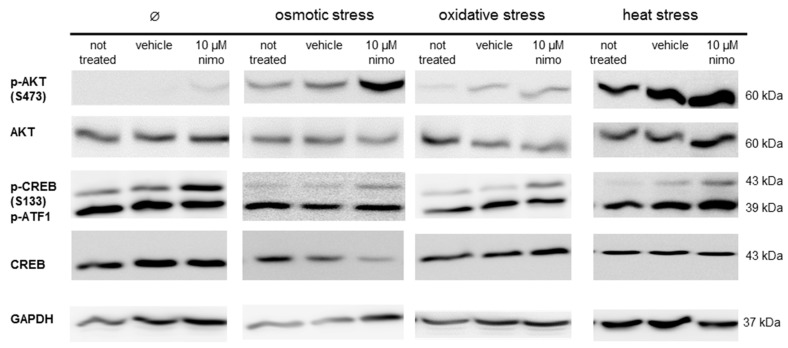
The immunoblot analysis of Schwann cells under different stress conditions. Schwann cells (SW10) were treated with 10 μM nimodipine (Nimo) or an equivalent volume of solvent (vehicle) and stressed (osmotic, oxidative or heat stress) or not stressed (∅) after 24 h. 50 μg of protein per sample was separated by SDS-PAGE and transferred onto a nitrocellulose membrane. The levels of AKT and CREB phosphorylation and the total protein amounts were detected with specific antibodies. The loading control served for the immunostaining of GAPDH. The figure shows a representative Western blot of three biological replicates.

**Figure 6 ijms-20-04578-f006:**
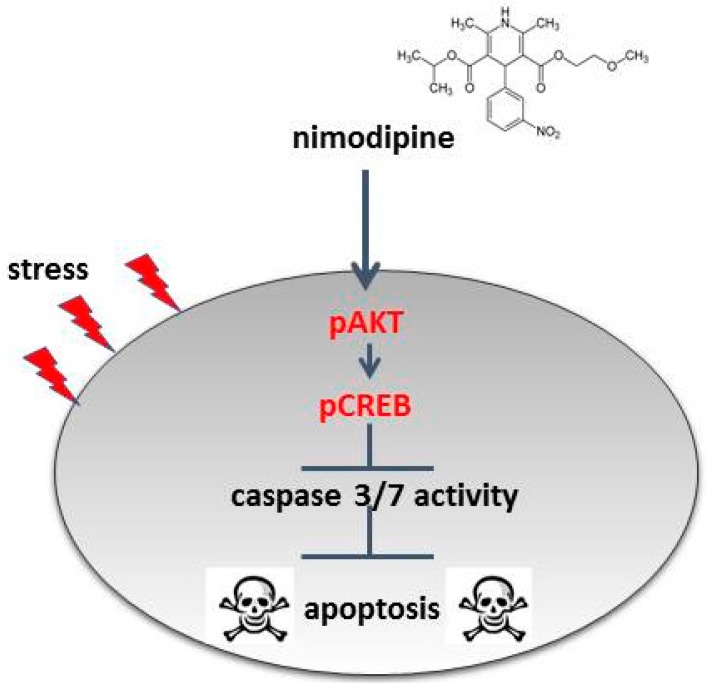
Model of the nimodipine neuroprotective mode of action. The nimodipine treatment leads to the activation of AKT and CREB signaling. AKT and CREB phosphorylation inhibits the activation of caspases and thereby the initiation of cell apoptosis.

**Figure 7 ijms-20-04578-f007:**
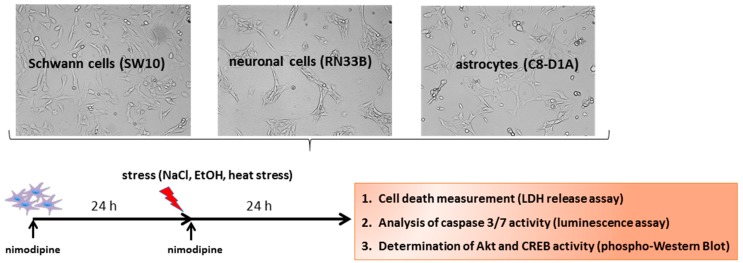
Scheme of the experimental procedure. To identify the neuroprotective molecular mechanisms of nimodipine, cell lines were pre-treated for 24 h with different concentrations of nimodipine. Then, the cells were incubated with 450 mM NaCl (osmotic stress), 2% EtOH (oxidative stress) or 6 h at 42 ° C (heat stress) and treated again with nimodipine for another 24 h. After that, cell death was measured via an LDH release assay, the caspase activation was analyzed via a Caspase 3/7-Glo assay and the cell signaling pathways were examined via Western blot.

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
