# Peer review of "Nimodipine-Dependent Protection of Schwann Cells, Astrocytes and Neuronal Cells from Osmotic, Oxidative and Heat Stress Is Associated with the Activation of AKT and CREB"

_ijms, 2019, doi:10.3390/ijms20184578_

Round 1
Reviewer 1 Report
The manuscript entitled “Nimodipine-dependent protection of Schwann cells, astrocytes and neuronal cells from osmotic, oxidative and heat stress is associated by activation of Akt and CREB” examines the effect of nimodipine in cell-subtype specific mouse cell culture. The calcium channel antagonist is generally found to be protective, reducing caspase activation, LDH release and increasing Akt and CREB. Generally, the methods are sound. The paper would benefit from careful proofreading to ensure subject/verb agreement and correct singular/plural grammar.
Presentation of the data could be made less laborious by planning meaningful comparisons, rather than comparing every single condition to each other. It is not truly meaningful. The statistical analysis is unclear. It is not surpising that a one way ANOVA would find significance when 4 conditions are included. Since the analysis is not described in detail, we do not know the magnitude of the group effect, as we are not given anything but p values. It is stated that data are presented as mean =/- SD. Is this correct? It is not SEM? The data are exceptionally tight for the SD to be represented on each bar. Are the data the average of each set of observations, for example, the methods state “Wells were measured at four points.”
The discussion handles the results as if all the drug were perfectly protecting at every dose in response to all the stressors. This is not the case and should be discussed. For instance, caspase activation is greatly increased in astrocytes at the highest dose of the drug when paired with heat stress. More discussion should center around what it means for each cell type. What is the significance of each cell type in recovery of function? As written it seems assumed that the reader knows how each cell might contribute. Indeed, reactive gliosis might be a negative consequence. Are the results necessarily indicative of protective?
Reviewer 2 Report
The authors indicate that the effect of nimodipine on various cell types of the nervous system has not been studied. The in vitro study analyzed the effect of nimodipine on Schwann cell, neuronal cell and astrocyte stress-induced cell death and associated molecular mechanisms. Cell lines from Schwann cells, neuronal cells and astrocytes were pretreated with nimodipine and incubated under stress conditions such as osmotic, oxidative and heat stress. The authors found decreased caspase activity and increased Akt and CREB phosphorylation could be observed after nimodipine treatment. The results indicate a cell type-independent neuroprotective effect of nimodipine treatment via activation of CREB and Akt signaling pathway and reduction of caspase 3 activity. The authors conclude that the cell type independent neuroprotection by nimodipine against a wide range of stress-induced neuronal damages during surgery or various neurodegenerative diseases.
Comment:
Akt and CREB interact with the nuclear receptor Sirtuin 1 (Sirt 1) that is critical to neuron survival. In recent studies Sirt 1 has been shown to be important to neuromuscular junction and terminal schwann cell survival. CREB/Sirt 1 is controlled at the transcriptional level by nutrient availability with Sirt 1 (deacetylase) critical to Akt activation and cell growth. Sirt 1 (heat shock gene) is regulated by temperature dysregulation and stress with protein aggregation associated with mitochondrial apoptosis.
Questions:
Q1. Is the effects of Nimodipine via oxidative and heat stress mediated via CREB-Sirt 1-Akt interactions in Schwann cells and neuron cells?
Q2. Can nutritional therapy (Sirt 1 activators versus inhibitors) determine the therapeutic use of Nimodipine in the treatment of neuromuscular decline and neurodegenerative disease?
RELEVANT REFERENCES:
Snyder-Warwick AK, Satoh A, Santosa KB, Imai SI, Jablonka-Shariff A. Hypothalamic Sirt1 protects terminal Schwann cells and neuromuscular junctions from age-related morphological changes. Aging Cell. 2018 Aug;17(4):e12776. Noriega LG, Feige JN, Canto C, Yamamoto H, Yu J, Herman MA, Mataki C, Kahn BB, Auwerx J. CREB and ChREBP oppositely regulate SIRT1 expression in response to energy availability. EMBO Rep. 2011 Sep 30;12(10):1069-76. Fusco S, Leone L, Barbati SA, Samengo D, Piacentini R, Maulucci G, Toietta G, Spinelli M2, McBurney M, Pani G, Grassi C. A CREB-Sirt1-Hes1 Circuitry Mediates Neural Stem Cell Response to Glucose Availability. Cell Rep. 2016 Feb 9;14(5):1195-1205. Vinodkumar B. Pillai, Nagalingam R. Sundaresan, and Mahesh P. Gupta. Regulation of Akt signaling by Sirtuins: Its implication in cardiac hypertrophy and aging. Circ Res. 2014 Jan 17; 114(2): 368–378. Martins IJ. Heat Shock Gene Inactivation and Protein Aggregation with Links to Chronic Diseases. Diseases. 2018, 6;39:1-5. Luo X, Tao L, Lin P, Mo X, Chen H. Extracellular heat shock protein 72 protects schwann cells from hydrogen peroxide-induced apoptosis. J Neurosci Res. 2012 Jun;90(6):1261-9.
Reviewer 3 Report
Authors investigated the role of Nimodipine treatment in vitro toxicity of different cell types. Results seem interesting and definitely add new information to the exciting literature. Few suggestions are given below
1) Section 2.1 to 2.3 is the effect of Nimodipine in different cell types during different stress condition. All having a similar experimental condition in different cell types. Please describe this result in one heading.
2) Section 2.4 and 2.5 are associated with a molecular mechanism with Nimodipine dependent protection. it needs more claim to confirm established this finding. Only 2 experiments, not enough. QPCR experiment with known genetic marker helpful here.
3) Add future perspective in conclusion.
